# Proposal of Evaluation Method for Crack Propagation Behaviors of Second-Generation Acrylic Adhesives under Mode I Static Loading

**DOI:** 10.3390/polym15081878

**Published:** 2023-04-14

**Authors:** Yuki Ogawa, Kimiyoshi Naito, Keisuke Harada, Hiroyuki Oguma

**Affiliations:** 1Polymer Matrix Hybrid Composite Materials Group, Research Center for Structural Materials, National Institute of Materials Science, 1-2-1 Sengen, Tsukuba 305-0047, Japan; 2Department of Mechanical Engineering, Graduate School of Engineering, Kobe University, 1-1 Rokkodai-cho, Nada-ku, Kobe 657-8501, Japan; 3Department of Aerospace Engineering, Tohoku University, 6-6-1 Aramaki Aza Aoba, Aoba-ku, Sendai 980-8579, Japan; 4Naval Platform and Signature Technology Division, Naval Systems Research Center, Acquisition, Technology & Logistics Agency, 2-2-1 Nakameguro, Meguro-ku, Tokyo 153-8630, Japan

**Keywords:** acrylic adhesive, crack propagation, fracture toughness, cohesive zone

## Abstract

Second-generation acrylic (SGA) adhesives, possessing high strength and toughness, are applicable in automotive body structures. Few studies have considered the fracture toughness of the SGA adhesives. This study entailed a comparative analysis of the critical separation energy for all three SGA adhesives and an examination of the mechanical properties of the bond. Loading-unloading test was performed to evaluate crack propagation behaviors. In the loading–unloading test of the SGA adhesive with high ductility, plastic deformation was observed in the steel adherends; the arrest load dominated the propagation and non-propagation of crack for adhesive. The critical separation energy of this adhesive was assessed by the arrest load. In contrast, for the SGA adhesives with high tensile strength and modulus, the load suddenly decreased during loading, and the steel adherend was not plastically deformed. The critical separation energies of these adhesives were assessed using the inelastic load. The critical separation energies for all the adhesives were higher for thicker adhesive. Particularly, the critical separation energies of the highly ductile adhesives were more affected by the adhesive thickness than highly strength adhesives. The critical separation energy from the analysis using the cohesive zone model agreed with the experimental results.

## 1. Introduction

The automotive industry involves lightweight and high-strength structural designs that consider environmental issues [1,2,3]. Conventional welding technologies have been extended to dissimilar materials [4,5]. Adhesive bonding is applied based on features, such as bonding for dissimilar materials, vibration resistance, and high work efficiency [6,7]. Among these, structural epoxy [8] and acrylic [9] adhesives, with both strength and toughness, are available for body structures in the automotive and aerospace fields. Particularly, second-generation acrylic (SGA) adhesives [10,11,12,13,14] have attracted considerable attention. The SGA adhesive is a two-component room-temperature-curable acrylic adhesive with acrylic resin and elastomer as the main constituents. SGA adhesives are cured to progress through radical reactions in the oxidation-reduction reaction in a short time. This is because a radical reaction with high reactivity occurs at low temperatures. A cured SGA adhesive exhibits a sea-island structure composed of hard acrylic resins dispersed in a soft elastomer. SGA adhesives have both high shear and tensile strengths, owing to the hard acrylic resins providing high impact resistance and peel strength compared to soft elastomers [15,16]. Thus, SGA adhesives offer disadvantages that are similar to structural epoxy adhesives. Furthermore, the changes in the mixing ratio of SGA adhesive do not affect its material characteristics.

Finite element method (FEM) analysis has been utilized for the evaluation of the strength of automobile body structures that use structural adhesives. In recent years, evaluation methods based on the cohesive zone model have been widely applied. The fracture toughnesses for Mode I and II loading are calculated using double cantilever beam (DCB) [17,18,19] and end notch fracture (ENF) [20,21] tests, respectively. The joint strength for both the fracture modes is predicted using the cohesive zone model [22,23,24]. However, studies that focus on the critical separation energy for the SGA adhesives in each fracture mode are remarkably limited. For adhesive under Mode I loading, we previously proposed an evaluation method of the fracture toughness [25]. In the DCB tests, the energy release rate cannot be correctly estimated if the plastic deformation occurs in thin steel adherends bonded with polyurethane adhesives. Therefore, a loading–unloading test was performed to evaluate the crack propagation behaviors of the adhesives even when plastic deformation occurs in thin adherends. The arrest load in the loading–unloading test was found to dominate the propagation and non-propagation of the crack of the adhesives. The arrest load did not affect the plastic deformation of the adherends. The adherends bonded with ductile adhesives confirmed large deformation in the loading–unloading test; furthermore, the large deformation of the adherends affected the fracture toughness. Thus, the critical separation energy, in which the Mode I-governed separation energy was determined using the arrest load, provided a safety evaluation in the design guidelines. However, the proposed method of fracture toughness can only be used to evaluate ductile adhesives such as polyurethane adhesives and is not suitable for examining structural acrylic adhesives.

To establish the design guidelines for automobile body structures, various factors dominating the adhesive joint strength must be examined. Particularly, the adhesive thickness has a significant effect on joint strength. Thin adhesives have been recommended for achieving high joint strength for many adhesives [26,27]. However, studies on the thickness of SGA adhesives are scarce [28,29]. A comprehensive investigation of the relation of the adhesive thickness and the fracture mode for SGA adhesives is required. The present study evaluated the crack propagation behaviors of three types of SGA adhesives. Additionally, the crack propagation behaviors of 0.3 mm and 3.0 mm thick adhesives were evaluated. Additionally, experimental investigations were conducted along with the FEM analysis using a cohesive zone model.

## 2. Experimental Methods

### 2.1. Materials

A low carbon steel SS400 plate was used as the adherend. The yield stress, maximum stress, and fracture strain of the SS400 plate are 245 MPa, 400–510 MPa, and 21%, respectively. The SGA adhesive employed three types of Metal Lock series (Cemedine Co., Ltd., Tokyo, Japan). Y618H was the first type belonging to the flexible lock category, while the second type was Metal Lock belonging to the multi-purpose category. Lastly, Y611 Black S was the high heat resistance type of Metal Lock series.

Tensile testing of each SGA adhesive was performed using 2 mm thick dog-bone-type specimens with a gauge length of 10 mm based on JIS K 6251 No. 8 [30]. Tensile testing was conducted at a crosshead displacement speed of 1.0 mm min^−1^. The strain was measured using a contactless video extensometer.

### 2.2. Specimens and Test Conditions of Loading–Unloading Test

Figure 1 shows the shape and dimensions of the DCB specimen based on JIS K 7086 [31]. To unify the surface condition in the adherends, the steel adherends were cut into specimens of dimension 200 mm × 25 mm, and then sandblasted with brown fused alumina abrasives. Subsequently, the steel adherends were degreased by acetone. A release film Wrightlon 5200 with a length and thickness 30 mm and 25 μm, respectively, was inserted in the bond line as an artificial crack. The adhesive thicknesses after using stainless steel spacers were 0.3 mm and 3.0 mm. The DCB specimen was cured for three days at room temperature, and the adhesive thickness of the DCB specimens was then measured using a microscope VHX-6000 (Keyence, Osaka, Japan).

The loading–unloading test [25,32] depicted in Figure 2 was conducted with a crosshead displacement speed of 1.0 mm min^−1^ using EZ-LX (Shimadzu Corporation, Kyoto, Japan). First, preload was applied to the DCB specimen, wherein an artificial crack was propagated until a length of approximately 40 mm was obtained to form a natural crack, after which the preload was removed. Then, the DCB specimen was loaded until the crack propagated further to a length of approximately 5 mm at point A, followed by an interval. In the case of the ductile adhesives as in our previous research [25], the stress and strain at the crack tip of the adhesives were released, and the crack propagation occurred gradually in the adhesives, as shown by the blue line in Figure 2a. This crack growth behavior is related to creep crack growth. Thereafter, the crack growth completely stopped. Therefore, the load at point B in Figure 2a is defined as the arrest load, *P*_arr_. However, for high-strength adhesives such as epoxy resins, the crack of adhesives might rapidly propagate up to point A, and the load drops rapidly up to point B, as shown by the blue line in Figure 2b [33]. Subsequently, the tests were paused, and the crack was arrested at the load at point B. The load at point B in Figure 2b is also defined as the arrest load, *P*_arr_. As shown in Figure 2c, the crack length was calculated as the linear distance from the loading point to the crack tip at point B. This crack length was selected because it eliminates the plastic deformation of the adherends and captures the actual phenomenon [25]. This linear distance is defined as the apparent crack length, *a*. The DCB specimen was sufficiently unloaded and then reloaded, as indicated by the points B–C–B in Figure 2. The compliance *C* (=(*δ*_B_*–δ*_C_)/(*P*_B_*–P*_C_)) was obtained from the linear relationship between points B and C. Subsequently, the load–displacement curve was observed to exhibit nonlinearity at point D in Figure 2. The load at point D is referred to as the inelastic load *P*_ine_. After reaching the maximum load, *P*_max_, at point E, the crack propagates stably or rapidly. Such loading and unloading cycles were performed until the DCB specimen fractured completely. At least three specimens were investigated for each adhesive thickness to check for repeatability.

The fracture toughness of the adhesives under Mode I static loading was measured based on the cantilever beam theory using loading–unloading tests [25,32]. The apparent crack length and cube root of the compliance have a linear relationship across all the adhesives and adhesive thicknesses [34]. This relation is defined by Equation (1); equivalently, the relation between the apparent crack length and compliance can be expressed by a cubic equation (Equation (2)). The energy release rate in the loading–unloading test is defined as the Mode I-dominated separation energy *SE*_I_, expressed by Equation (3). For ductile adhesives, the arrest load eliminates the plastic deformation of the adherends. It only dominated the propagation and non-propagation of crack for adhesive in the loading–unloading test [25,32]. The critical separation energy *SE*_IC_ of these adhesives is referred to as the Mode I-dominated separation energy by the arrest load in Equation (4) [25,32]. However, for the adhesives in the case of rapid crack growth, the arrest load was very small. Moreover, the crack of adhesives did not regrow at the arrest load when they propagated rapidly. However, the crack in this adhesive repropagated near the inelastic load during reloading. In other words, the inelastic load that governs the crack propagation behavior is desirable for evaluating the critical separation energy for this type of adhesive. Therefore, critical separation energy for rapid crack growth is defined separately as the critical separation energy in rapid crack growth, *SE*_IC-rapid_, evaluated using the inelastic load, as shown in Equation (5).
(1)$$\sqrt[{3}]{C}=A_{1}a+A_{2}$$
(2)$$C=A_{1}^{3}a^{3}+3A_{1}^{2}A_{2}a^{2}+3A_{1}A_{2}^{2}a+A_{2}^{3}$$
(3)$$SE_{I}=\frac{P^{2}}{2B}\frac{dc}{da}=\frac{P^{2}}{2B}\left(3A_{1}^{3}a^{2}+6A_{1}^{2}A_{2}a+3A_{1}A_{2}^{2}\right)$$
(4)$$SE_{\mathrm{IC}}=\frac{P_{\mathrm{arr}}^{2}}{2B}\frac{dc}{da}=\frac{P_{\mathrm{arr}}^{2}}{2B}\left(3A_{1}^{3}a^{2}+6A_{1}^{2}A_{2}a+3A_{1}A_{2}^{2}\right)$$
(5)$$SE_{\mathrm{IC}-\mathrm{rapid}}=\frac{P_{\mathrm{ine}}^{2}}{2B}\frac{dc}{da}=\frac{P_{\mathrm{ine}}^{2}}{2B}\left(3A_{1}^{3}a^{2}+6A_{1}^{2}A_{2}a+3A_{1}A_{2}^{2}\right)$$

### 2.3. FEM Analysis

To examine the validity of the critical separation energy of the SGA adhesives, a FEM analysis was carried out using ABAQUS. Figure 3 shows the boundary conditions of the analysis model for the DCB specimen. The analysis was performed using a two-dimensional symmetric model that incorporates iso-parametric quadrilateral elements. The material properties of the adherends were elastic materials. The arrest load for evaluating the crack propagation behaviors of the SGA adhesives was not influenced by the plastic deformation of adherends. Therefore, the FEM analysis does not consider the plastic deformation of adherends. The material properties of the adhesives were similar to those of an elastoplastic material using the tensile test results. A cohesive zone was introduced at the boundary between the adhesive layer and the ultrathin elastic using the two-node linear beam elements. A pre-crack of 20 mm was introduced from the bonded edge. This analysis calculated the adhesive thicknesses of 0.1, 0.3, 1.0, and 3.0 mm. The number of nodes and elements was the same for the adhesives. The node and element numbers for 0.1 mm were 735 and 480, respectively. Similarly, adhesives with a thickness of 0.3 mm, 1.0 mm, and 3.0 mm consisted of 919, 919, and 948 nodes, and 662, 662, and 990 elements, respectively. The cohesive parameter was used with the zero-thickness based on the linear relationship between each critical separation energy and the adhesive thickness. In the analysis boundary conditions, an elastic bar was fixed in every direction, the load point was coupled to the same position as the left corner of the upper adherend, and the displacement at the loading point was 6 mm. In the FEM analysis, the viscoelastic term of the adhesives was not defined, and the critical separation energy was calculated using the analysis results corresponding to the inelastic load for all the adhesives.

## 3. Results and Discussion

### 3.1. Tensile Testing of SGA Adhesive

Figure 4 shows the stress–strain (*σ*–*ε*) curves obtained using the tensile tests of each adhesive. The black, blue, and red lines indicate the tensile test results of the Y618H, Metal Lock, and Y611 Black S adhesive, respectively. At least three specimens of each adhesive were tested to check for repeatability. Table 1 presents the mean and standard deviation values of the tensile test results in parentheses. The adhesive with the highest tensile strength and tensile modulus was the Y611 Black S adhesive. However, it offered the lowest elongation. The tensile strength and elongation of the Metal Lock adhesive were similar to those of the Y611 Black S adhesive. Contrastingly, the Y618H adhesive had the lowest tensile strength but the greatest elongation. The Y611 Black S and Metal Lock adhesives had high tensile strengths and moduli, while the Y618H adhesive exhibited high ductility.

### 3.2. Loading–Unloading Tests

Figure 5 shows the load–displacement curves of the loading–unloading tests for each adhesive. The arrest and inelastic loads are plotted in Figure 5. Figure 6 shows the relationship between the Mode I-dominated separation energy and increment of the apparent crack length, for all SGA adhesives and adhesive thicknesses. In the results of the loading–unloading tests for the Y618H adhesives that are presented in Figure 5a,b, stable crack propagation was observed for 0.3 mm as well as 3.0 mm thick adhesive. As shown in the figure, the crack of the Y618H adhesive propagated near the arrest load. The crack began to repropagate near the arrest load, as indicated by Figure 6a,b. For the Y618H adhesives, the arrest load governed the crack regrowth. This observation is similar to the ductile adhesives observed in our previous research [25]. In the case of Metal Lock adhesives, the crack propagation for a 0.3 mm thick adhesive exhibited a stable behavior during the test interval, as shown in Figure 5c. However, for a 3.0 mm thick adhesive as shown in Figure 5d, stable and rapid crack propagation with a sudden decrease in load was observed in one specimen. That is, the crack propagation behaviors of the Metal Lock adhesives, in the loading–unloading tests depend on the adhesive thickness. For the Y611 Black S adhesives (as shown in Figure 5e,f), the load–displacement curve exhibited nonlinearity. A sudden decrease in the load was observed during loading, while the arrest load was insignificant, irrespective of the adhesive thickness. The cracks in the adhesives propagated rapidly in the case of both adhesive thicknesses. Moreover, in the relationship between the Mode I-dominated separation energy and increment of the apparent crack length for the Metal Lock (Figure 6c,d) and Y611 Black S (Figure 6e,f) adhesives, the arrest load was considerably small, and the cracks of the adhesives did not repropagate at the arrest load. The crack propagation behaviors for Metal Lock and Y611 Black S adhesives govern the inelastic load.

The loading–unloading test results were examined in regard to the actual crack propagation behavior. Figure 7 shows the observation results from the side of DCB specimen before resuming the test, when the crack propagation reached approximately 70 mm for each adhesive and adhesive thickness. Figure 8 shows the representative fracture surfaces of each adhesive and adhesive thickness. In all the sub-figures of Figure 8, the areas of stable and rapid crack propagation are represented by red and black arrows, respectively, in the loading–unloading tests. For the Y618H adhesive, multiple voids were formed inside the adhesives beyond the crack tip in the crack propagation direction, and cracks in the adhesives propagated along the voids, as shown in Figure 7a,b. The crack within a thin adhesive in Figure 7a propagated through the center of the adhesive. For a thicker adhesive, the crack propagation of the adhesive occurred in near the steel adherends, as shown in Figure 7b. This crack propagation behavior was similar to that observed in other ductile SGA adhesives [25]. The fracture surface of the Y618H adhesive was a mainly cohesive failure, for both adhesive thicknesses as shown in Figure 8a,b. The plastic deformation was observed in the steel adherends. Because the crack propagation behaviors of the Y618H adhesives demonstrated stability in the load–displacement curves, it is preferable to evaluate the critical separation energy by the arrest load with the plastic deformation in the steel adherends. Conversely, the crack of the Metal Lock adhesives propagated through the center of the adhesive, irrespective of the adhesive thickness, as shown in Figure 7c,d. The fracture surface of the Metal Lock adhesives for 0.3 mm thick adhesive shows an overall cohesive failure as presented in Figure 8c. In the case of the 3.0 mm thick Metal Lock adhesives, as shown in Figure 8d, a part of the fracture surface showed stable crack propagation owing to the cohesive failure in the white area. However, most of the fracture surfaces confirmed the stick-slip phenomenon because of unstable brittle fractures in the black area. The crack of adhesives did not propagate in the critical separation energy by the arrest load, as aforementioned. Therefore, the evaluation using the arrest load was conservative. The plastic deformation did not occur in the steel adherends regardless of the adhesive thickness. Hence, the critical separation energy of the Metal Lock adhesives must be selected for evaluation by the inelastic load to capture the actual phenomena. The cracks in the Y611 Black S adhesives propagated in the center of the adhesive layer as well as the Metal Lock adhesives, as shown in Figure 7e,f. The fracture surfaces of the Y611 Black S adhesive shown in Figure 8e,f represent an overall stick-slip phenomenon and part of the white area under load conditions in the loading–unloading tests, regardless of the adhesive thickness. This is because of the adhesives with high strength and low elongation, such as the modified epoxy resin. The plastic deformation was not observed in the steel adherends of the DCB specimen of the Y611 Black S adhesive. It is assumed that the inelastic load in the load–displacement curves show the crack propagation behaviors of the adhesives. Therefore, to evaluate the crack propagation behaviors of the Y611 Black S adhesives under Mode I static loading, the critical separation energy in rapid crack propagation calculated using the inelastic load must be used.

Figure 9 depicts the relationship between the apparent crack length and the cube root of the compliance. The green curve in Figure 9 indicates these relationships based on the cantilever beam theory. There was a linear relationship between the apparent crack length and the cube root of the compliance at each adhesive as well as adhesives of varying thicknesses. Figure 10 shows the relations of the critical separation energy and apparent crack length for each adhesive and its different thicknesses. As aforementioned, the critical separation energy of the Y618H adhesives was evaluated using the arrest load, and the corresponding values for the Metal Lock and Y611 Black S adhesives were calculated using the inelastic load.

The critical separation energy has a similar value for every apparent crack length irrespective of the adhesive thickness. Figure 10 and Table 2 show the mean values of the critical separation energy for each adhesive of different thicknesses. The standard deviation of the results of loading–unloading tests is presented in parentheses in Table 2. Additionally, Table 2 shows the zero-thickness critical separation energy calculated using the linear relations of the critical separation energy and adhesive thickness. The critical separation energy of all the adhesives increased with increasing adhesive thickness. Particularly, compared to other adhesives, the critical separation energy of the Y618H adhesive was more influenced by the adhesive thickness. The effect of adhesive thickness in terms of fracture toughness and joint strength, the ductile adhesives are considered to be more affected by the adhesive thickness than brittle adhesive [27]. As shown in Table 1, the Y618H adhesive is a highly ductile adhesive with lower strength and lower elasticity but higher breaking strain than the Y611 Black S and Metal Lock adhesives. Therefore, the critical separation energy of the Y618H adhesive was considered to be most affected by the adhesive thickness.

### 3.3. FEM Analysis

The critical normal separation curves were determined from the tensile test results of the adhesives shown in Figure 1. The normal separation values of Y618H, Metal Lock, and Y611 Black S adhesives are 0.29 mm, 0.038 mm, and 0.058 mm, respectively. In the previous section, the critical separation energy for the three types of SGA adhesives was proposed based on crack propagation behaviors. To study whether the critical separation energy of the adhesives is appropriate, FEM analysis was carried out using the zero-thickness triangular cohesive elements. Figure 11 shows the experimental and analysis results of the relations of the critical separation energy and the adhesive thickness. The analysis results of the critical separation energy for each adhesive were in good agreement with the experiments. This indicates that the arrest load dominates the propagation and non-propagation of crack for highly ductile adhesive, such as Y618H, as observed in the previous research [25]. However, for high-strength SGA adhesives, such as the Metal Lock and Y611 Black S adhesives, the inelastic load dominated the crack propagation. In other words, the crack propagation behaviors under Mode I loading of various SGA adhesives could be evaluated using the loading–unloading tests by selecting the load in this test according to the crack propagation behaviors and mechanical properties of the SGA adhesives.

## 4. Conclusions

To evaluate Mode I crack propagation behaviors of the SGA adhesives, the fracture toughness under Mode I loading was calculated in a loading–unloading test. For the Y618H adhesive, plastic deformation was observed in the steel adherends, and the critical separation energy of the Y618H adhesive was assessed using the arrest load. Moreover, the critical separation energy of Y618H adhesive for 0.3 mm and 3.0 mm thicknesses were 499 J m^−2^ for and 1456 J m^−2^, respectively. In the case of the Metal Lock and Y611 Black S adhesives, the load suddenly decreased during loading, and plastic deformation was not observed in the steel adherend. Thus, the cracks in the adhesives propagated rapidly in the adhesives of different thicknesses. The critical separation energy in rapid crack propagation calculated using the inelastic load must be used to evaluate the crack propagation behaviors of the Metal Lock and Y611 Black S adhesives. For the Metal Lock adhesive, the critical separation energy in rapid crack propagation was 673 J m^−2^ and 918 J m^−2^ for 0.3 mm and 3.0 mm thicknesses, respectively. For the Y611 Black S adhesive, the critical separation energy in rapid crack propagation was 673 J m^−2^ and 918 J m^−2^ for 0.3 mm and 3.0 mm thicknesses, respectively. The critical separation energy of all the adhesives increased with increasing adhesive thickness. In particular, the critical separation energy of the Y618H adhesives (low strength, low tensile modulus, and high breaking strain in a bulk specimen) was more affected by the adhesive thickness than other adhesives with high strength, high tensile modulus, and low breaking strain in a bulk specimen. The experimental results agreed with the analysis results by zero-thickness triangular cohesive elements. Therefore, the Mode I crack propagation behaviors of various types of SGA adhesives can be evaluated in the loading–unloading tests.

## Figures and Tables

**Figure 1 polymers-15-01878-f001:**
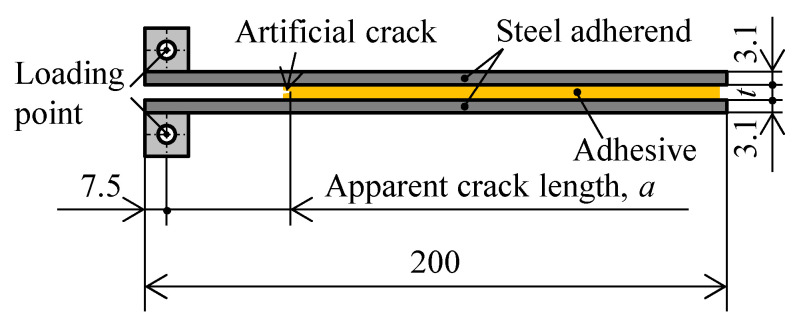
Shape and dimensions of the double cantilever beam (DCB) specimen (unit: mm).

**Figure 2 polymers-15-01878-f002:**
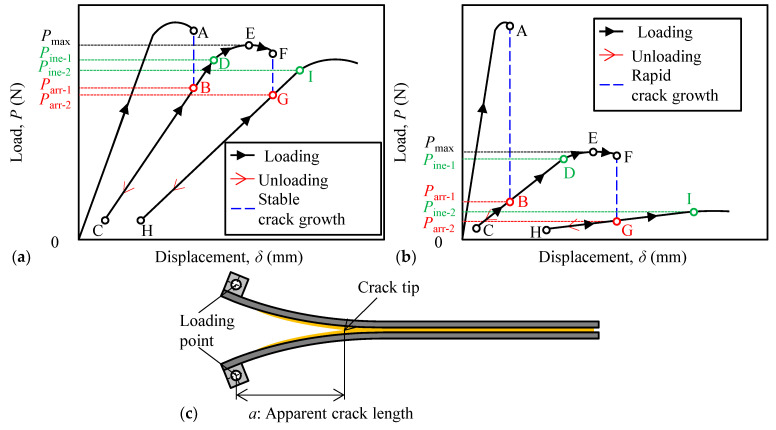
Illustration of loading–unloading test method in case of (**a**) stable crack propagation and (**b**) rapid crack propagation; (**c**) Schematic illustration on the definition of apparent crack length.

**Figure 3 polymers-15-01878-f003:**
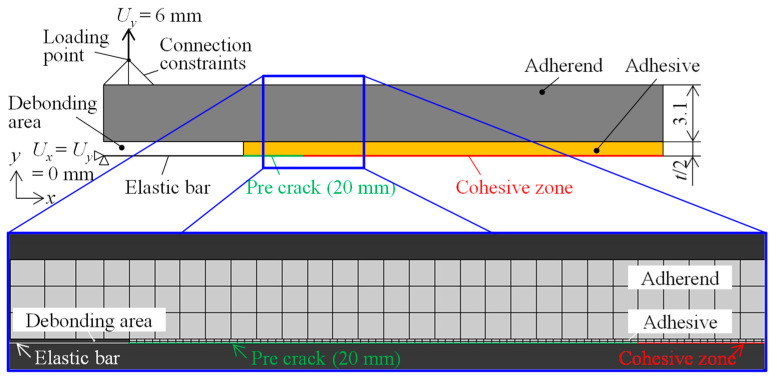
Finite element analysis model for a DCB specimen.

**Figure 4 polymers-15-01878-f004:**
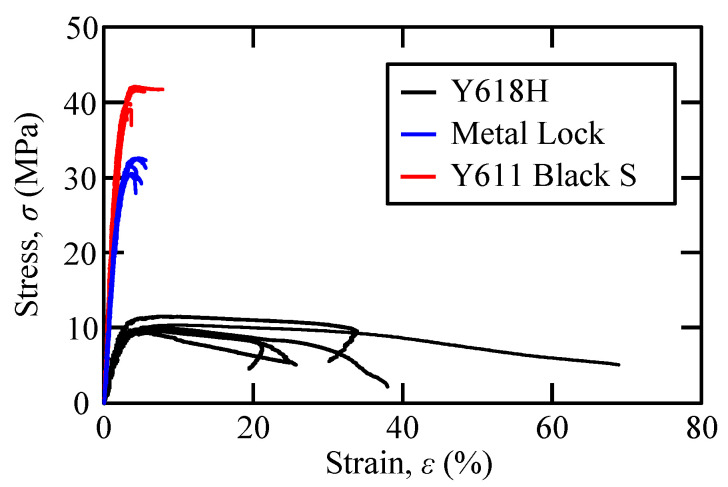
Stress–strain curves of Y618H, Metal Lock, and Y611 Black S adhesive.

**Figure 5 polymers-15-01878-f005:**
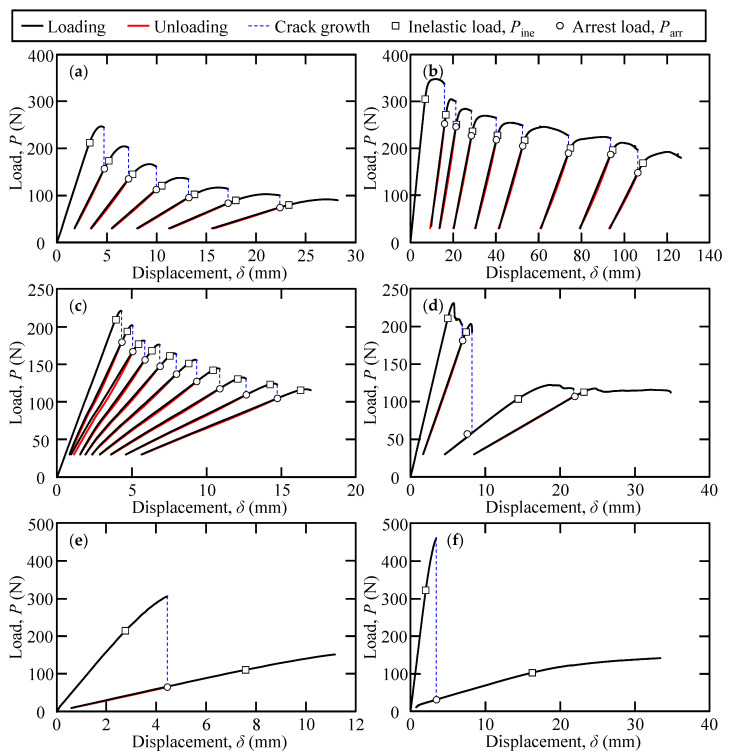
Load–displacement curves of the loading–unloading tests of (**a**) 0.3 mm thickness of Y618H, (**b**) 3.0 mm thickness of Y618H, (**c**) 0.3 mm thickness of Metal Lock, (**d**) 3.0 mm thickness of Metal Lock, (**e**) 0.3 mm thickness of Y611 Black S, and (**f**) 3.0 mm thickness of Y611 Black S.

**Figure 6 polymers-15-01878-f006:**
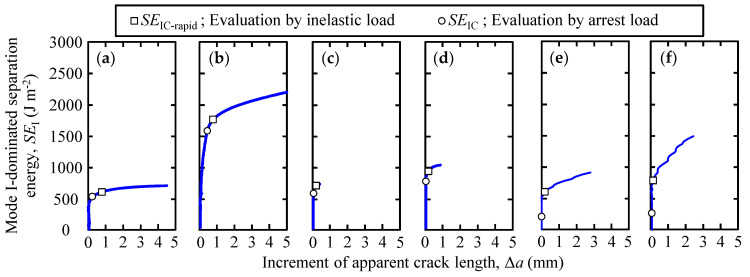
Relationship between the Mode I-dominated separation energy and increment of apparent crack length of (**a**) 0.3 mm thickness of Y618H, (**b**) 3.0 mm thickness of Y618H, (**c**) 0.3 mm thickness of Metal Lock, (**d**) 3.0 mm thickness of Metal Lock, (**e**) 0.3 mm thickness of Y611 Black S, and (**f**) 3.0 mm thickness of Y611 Black S.

**Figure 7 polymers-15-01878-f007:**
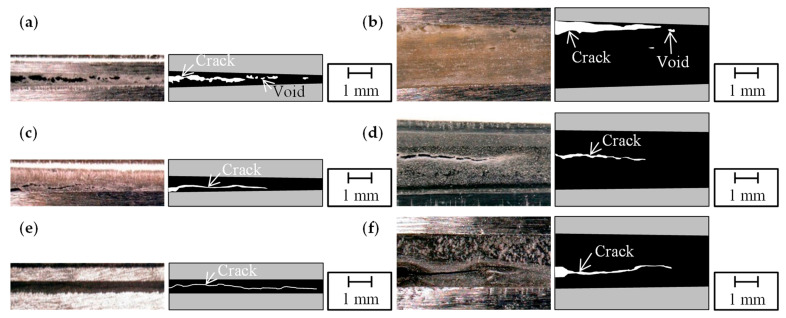
Observation results from the side of DCB specimen before resuming the test of (**a**) 0.3 mm thickness of Y618H, (**b**) 3.0 mm thickness of Y618H, (**c**) 0.3 mm thickness of Metal Lock, (**d**) 3.0 mm thickness of Metal Lock, (**e**) 0.3 mm thickness of Y611 Black S, and (**f**) 3.0 mm thickness of Y611 Black S.

**Figure 8 polymers-15-01878-f008:**
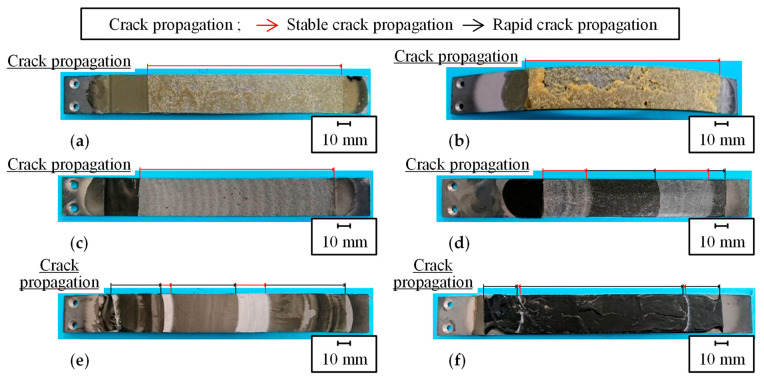
Fracture surfaces of the DCB specimen of (**a**) 0.3 mm thickness of Y618H, (**b**) 3.0 mm thickness of Y618H, (**c**) 0.3 mm thickness of Metal Lock, (**d**) 3.0 mm thickness of Metal Lock, (**e**) 0.3 mm thickness of Y611 Black S, and (**f**) 3.0 mm thickness of Y611 Black S.

**Figure 9 polymers-15-01878-f009:**
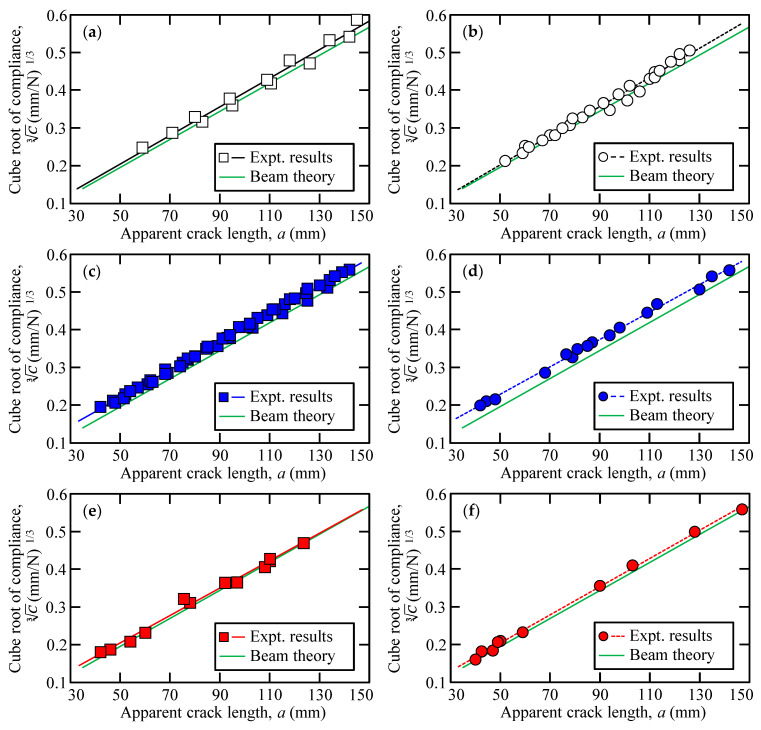
Relationship between the apparent crack length and the cube root of compliance of (**a**) 0.3 mm thickness of Y618H, (**b**) 3.0 mm thickness of Y618H, (**c**) 0.3 mm thickness of Metal Lock, (**d**) 3.0 mm thickness of Metal Lock, (**e**) 0.3 mm thickness of Y611 Black S, and (**f**) 3.0 mm thickness of Y611 Black S.

**Figure 10 polymers-15-01878-f010:**
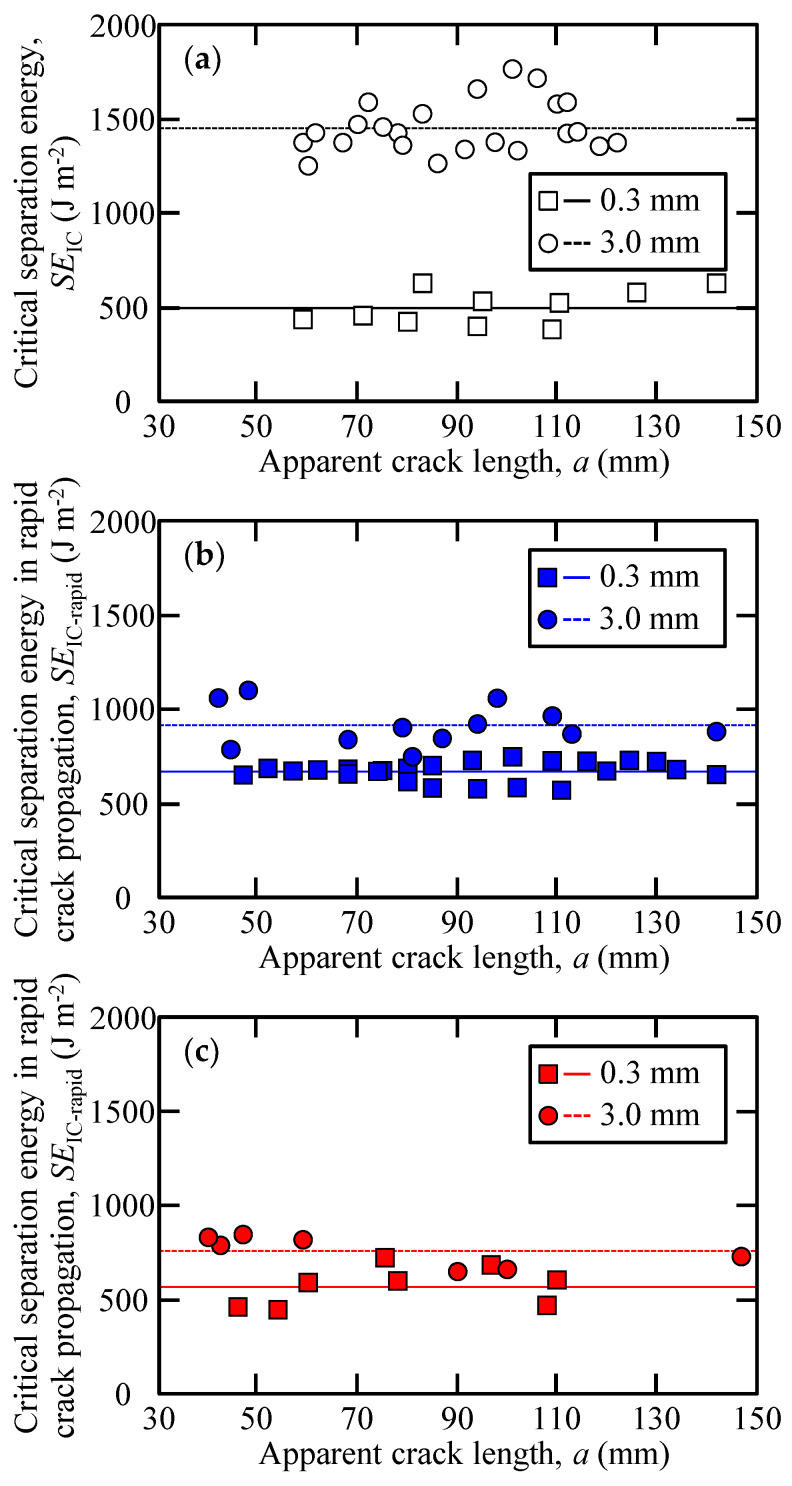
Relations of the critical separation energy and apparent crack length of (**a**) Y618H, (**b**) Metal Lock, and (**c**) Y611 Black S adhesive.

**Figure 11 polymers-15-01878-f011:**
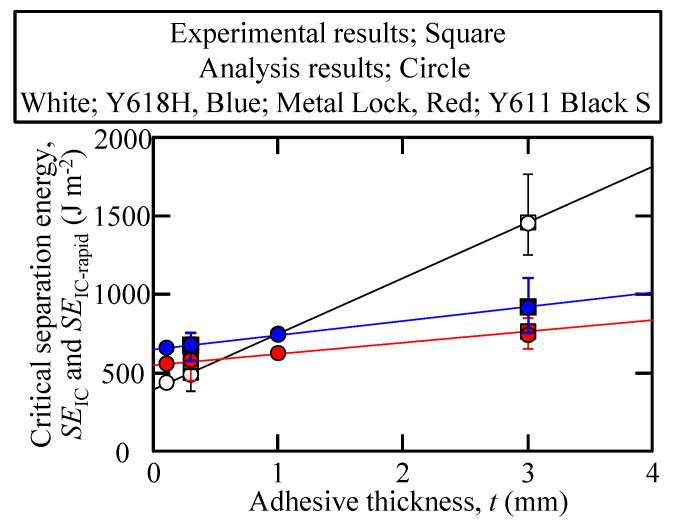
Analysis results showing the variation of the critical separation energy with respect to the adhesive thickness.

**Table 1 polymers-15-01878-t001:** Mechanical properties of Y618H, Metal Lock, and Y611 Black S adhesive (standard deviation is mentioned in parentheses).

Adhesive	Tensile Strength (MPa)	Tensile Modulus (MPa)	Strain at Failure (%)
Y618H	10.2 (0.7)	434 (37)	34.2 (16.3)
Metal Lock	31.7 (1.1)	1527 (120)	5.7 (2.3)
Y611 Black S	40.1 (1.7)	2017 (226)	4.6 (2.0)

**Table 2 polymers-15-01878-t002:** Mean values of the critical separation energy for Y618H, Metal Lock, and Y611 Black S adhesive of different thicknesses (standard deviation is mentioned in parentheses).

Adhesive	Critical Separation Energy, *SE*_IC_ or *SE*_IC-rapid_ (J m^−2^)		
	Zero Thickness	0.3 mm	3.0 mm
Y618H (*SE*_IC_)	393	499 (92)	1456 (138)
Metal Lock (*SE*_IC-rapid_)	646	673 (51)	918 (111)
Y611 Black S (*SE*_IC-rapid_)	547	568 (99)	761 (81)

## Data Availability

The authors declare that they have no known competing financial interests or personal relationships that could have appeared to influence the work reported in this paper.

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
