# Peer review of "Proposal of Evaluation Method for Crack Propagation Behaviors of Second-Generation Acrylic Adhesives under Mode I Static Loading"

_polymers, 2023, doi:10.3390/polym15081878_

Round 1

Reviewer 1 Report

The manuscript is interesting and suitable for Polymers. Authors must make a few changes before publishing the manuscript:

- ABSTRACT: Background, Methods, Results and Conclusion. Please respect the structure.

- Figures 2, 3 and 4 can be found in the previous article (https://doi.org/10.1016/j.ijadhadh.2022.103172). I think they are not necessary, you can just include the reference.

- The title does not correspond to the reference [29].

- Row 189: Why is 919 and 662 written twice?

- Please use the same scale on the y-axis (Load, P(N)) in Figure 5.

Reviewer 2 Report

The manuscript is well written. The figures are of acceptable quality and well supported by appropriate discussions. There are a few minor revisions that need to be made before the manuscript can be accepted for publication in Polymers:

1- In line 16, please define SGA. Each acronym and abbreviation should be defined in the first place in which they appear.

2- The introduction is too general. There is a great deal of difficulty in understanding the concept of this work, previous attempts and accomplishments, the remaining challenges, and the ideas of this work to resolve these issues. It is recommended that the authors rewrite the introduction section and define their critical solutions in a concise manner.

3- Figures 1-4 should be included in the Results and Discussion section (section 3), not in the Experimental section. In Section 2, the details of the experiments and instruments used in this study should be presented solely.

4- Line 115: Please describe the details of the microscope.
